# IDH Signalling Pathway in Cholangiocarcinoma: From Biological Rationale to Therapeutic Targeting

**DOI:** 10.3390/cancers12113310

**Published:** 2020-11-09

**Authors:** Massimiliano Salati, Francesco Caputo, Cinzia Baldessari, Barbara Galassi, Francesco Grossi, Massimo Dominici, Michele Ghidini

**Affiliations:** 1Division of Oncology, Department of Oncology and Hematology, University Hospital of Modena, 41125 Modena, Italy; francesco1990.caputo@libero.it (F.C.); cinzia.baldessari@libero.it (C.B.); massimo.dominici@unimore.it (M.D.); 2PhD Program Clinical and Experimental Medicine, University of Modena and Reggio Emilia, 41125 Modena, Italy; 3Division of Medical Oncology, Fondazione IRCCS Ca’ Granda, Ospedale Maggiore Policlinico, 20122 Milan, Italy; barbara.galassi@policlinico.mi.it (B.G.); francesco.grossi@policlinico.mi.it (F.G.); michele.ghidini@policlinico.mi.it (M.G.)

**Keywords:** biliary cancer, cholangiocarcinoma, gallbladder cancer, IDH, targeted therapy, precision medicine, ivosidenib

## Abstract

**Simple Summary:**

Cholangiocarcinoma is among the most challenging cancers to treat, associated with poor prognosis both in the early and advanced setting. In the last decade, a deeper understanding of disease biology and cholangiocarcinogenesis has led to an increasing awareness of the molecular heterogeneity underpinning this disease and to the identification of several vulnerabilities to be targeted. To this end, the therapeutic exploitation of IDH mutations is the first successful example of precision medicine in cholangiocarcinoma. In the ClarIDHy trial, the small molecule inhibitor Ivosidenib provided a survival advantage in pretreated patients with IDH1-mutant cholangiocarcinoma, thus expanded the armamentarium against this molecular subtype of disease.

**Abstract:**

Biliary tract cancers are anatomically distinct and genetically diverse tumors, evenly characterized by poor response to standard treatments and a bleak outlook. The advent of comprehensive genomic profiling using next-generation sequencing has unveiled a plethora of potentially actionable aberrations, changing the view of biliary tract cancers from an “orphan” to a “target-rich” disease. Recently, mutations in isocitrate dehydrogenase genes (IDH1/2) and fusions of the fibroblast growth factor receptor have emerged as the most amenable to molecularly targeted inhibition, with several compounds actively investigated in advanced-phase clinical trials. Specifically, the IDH1 inhibitor ivosidenib has been the first targeted agent to show a survival benefit in a randomized phase III trial of cholangiocarcinoma patients harboring IDH1 mutations. In this review article, we will focus on the IDH1/IDH2 pathway, discussing the preclinical rationale of its targeting as well as the promises and challenges of the clinical development of IDH inhibitors in biliary tract cancers.

## 1. Introduction

Biliary tract cancers (BTCs) consist of a heterogeneous group of aggressive malignancies arising from different locations of the biliary tree within and outside the liver. Based on the updated anatomical classification, BTCs encompass intrahepatic cholangiocarcinoma (iCCA), extrahepatic cholangiocarcinoma (eCCA) (further divided into perihilar (pCCA) and distal cholangiocarcinoma (dCCA)), gallbladder cancer (GBC), and ampulla of Vater cancer (AVC), also reflecting differences in epidemiology, aetiology, biology, prognosis, and therapeutic management (Figure 1) [1]. BTC is the second most common primary liver cancer after hepatocellular carcinoma and constitutes approximately 3% of all gastrointestinal tumors [2]. Although considered a relatively rare entity, the overall incidence of BTC has been steadily rising in the last decades, mainly as a result of improved diagnostic capabilities and changes in disease classification [3]. Despite recent advances, the prognosis of BTC is still meager. Indeed, only 10−20% of cases are amenable to curative-intent surgery, and, even in resected cases, the 5-year overall survival is less than 50%. On the other hand, the vast majority of patients diagnosed with unresectable advanced disease are candidates to palliative chemotherapy aimed at prolonging survival and maintaining an acceptable quality of life [4]. The combination of cisplatin and gemcitabine is the standard-of-care first-line treatment based on the results of ABC-02 and BT22 trials with a median overall survival (OS) inferior to 12 months [5,6]. Regarding later lines treatment, determining a role for chemotherapy is a recent achievement, with the results of the ABC-06 trial showing an advantage for the mFOLFOX6 regimen compared to active symptoms control, while beyond second-line, no high-level evidence currently supports the use of systemic treatment in clinical practice [7]. 

Lately, the advent of massive parallel sequencing technologies has enabled an in-depth understanding of the molecular landscape of BTC, unraveling several genomic vulnerabilities affecting metabolic, mitogenic, chromatin remodeling, and DNA repair signaling pathways [8,9].

Specifically, tumor profiling studies have reported that nearly 40% of BTCs harbor potentially actionable aberrations, among which are isocitrate dehydrogenase (IDH) 1/2 mutations (10%), fibroblast growth factor receptor (FGFR) fusions (10%), HER2 amplifications/mutations (10−15%), BRAF^V600E^ mutation (3%), BRCA2 mutations (3%), and microsatellite instability (1%) [8,9]. To this end, IDH 1/2 mutations and FGFR 2 fusions are being clinically exploited as the most relevant therapeutic targets so far, with several targeted agents showing unprecedented results in refractory disease settings [10,11]. In this article, we focused on the aberrant IDH signaling pathway, reviewing its biological relevance in cancer together with the preclinical and clinical development of selective IDH inhibitors as well as future perspectives in cholangiocarcinoma.

## 2. The Genomic Landscape of Cholangiocarcinoma

From a mutational standpoint, cholangiocarcinoma lies in the middle of the spectrum of malignancies, with roughly the same genomic burden of aberrations between iCCA and eCCA, showing a median of 39 and 35 non-synonymous mutations per tumor, respectively [12]. In recent years, integrative profiling studies have started disentangling the complex molecular landscape underpinning CCA, thereby shedding initial light on the biological heterogeneity across anatomical subtypes. To this end, IDH1/2 mutations (4.9–36%), FGFR 1–3 fusions, mutations and amplifications (11–45%), as well as BAP-1 mutations (13%) have been reported to occur more frequently in iCCA, whereas KRAS mutations (8.3–42%), SMAD4 mutations (21%), and ERBB2/3 amplifications (11–17%) have been observed more commonly in eCCA [13,14,15]. Moreover, Nakamura and colleagues described how eCCA was more specifically associated with previously unknown aberrations, such as ATP1B-PRKACA and ATP1B-PRKACB fusions, along with mutations in ELF3 and ARID1B genes [16]. 

In addition, cholangiocarcinoma also displays genomic diversity according to aetiological risk factor as demonstrated by the higher mutational burden found in liver fluke–driven tumors (median 4700 vs 3143 somatic mutations/tumor) and the enrichment for ERBB2 amplification and TP53 mutations. In contrast, non-liver fluke-associated cholangiocarcinoma has been shown to harbor high copy-number aberrations, PD-1/PD-L1 expression, epigenetic mutations involving IDH1/2 and BAP-1, and FGFR/PRKA-related gene rearrangement [8,17]. Based on the genomic complexity of cholangiocarcinoma with high molecular heterogeneity and multiple deranged oncogenic networks implicated, several efforts have been headed to subtype this disease at the molecular level in order to obtain clinically relevant information to be exploited. One of such attempts was pursued by the International Cancer Genome Consortium using a multiplatform approach on 489 cases coming from 10 different countries [8]. In this study, cholangiocarcinomas were stratified in four molecular subsets (clusters 1 to 4), each characterized by distinct genomic, epigenomic, and clinico-pathological features and a different prognostic impact. Interestingly, clusters 3 and 4 were associated with better survival than clusters 1 and 2. This study provided an in-depth molecular characterization that went beyond the site of origin of CCA as, for instance, the molecular clustering was replicated within each anatomical site separately. Another study by The Cancer Genome Atlas employed an integrative approach looking at somatic mutations, DNA methylation patterns, copy number alterations, and RNA expressions in a series of iCCA-predominant tumors [18]. This study identified the presence of a distinct subtype of IDH-mutant cholangiocarcinoma displaying upregulation of mitochondrial genes and DNA copy number variations and downregulation of chromatin modifier genes.

## 3. IDH Signaling Pathway in Cancer

Isocitrate dehydrogenase is an essential metabolic enzyme for cellular respiration in the tricarboxylic acid cycle. There are three main subtypes of IDH, with IDH1 and IDH2 being the most relevant for catalyzing the NADP^+^-dependent oxidative decarboxylation of isocitrate to α-ketoglutarate (α-KG) and CO_2_. IDH1 is localized in peroxisomes and cytosol, while IDH2 localizes to the mitochondria [19]. Recurrent somatic mutations usually occur at a single amino acid residue of both IDH1 (arginine 132) and IDH2 (arginine 172 or arginine 140) [20,21]. IDH mutations are considered gain-of-function and lead to the disruption of the normal catalytic activity of IDH1/2, ultimately resulting in increased conversion of α-KG to D-2-hydroxyglutarate (D-2HG), which acts as an oncometabolite, promoting tumor proliferation and metastasis development through several pathways, such as DNA methylation and activation of VEGFR [19,22,23] (Figure 2). Levels of 2-HG were found to be significantly higher in IDH1-mutant glioma and acute myeloid leukemia cell lines than wild-type, causing epigenetic dysfunction and inducing a DNA hypermethylation phenotype [22,23,24,25]. Furthermore, the D-2HG-induced dysregulation of histone and DNA methylation inhibited normal cellular differentiation, promoting malignant transformation [26,27,28]. IDH1-mutant creates a heterodimer with wild-type IDH1, silencing the wild-type activity and decreasing α-KG levels. Lower levels of α-KG can inhibit the degradation of hypoxia-inducible factor 1α (HIF-1α) and enhance angiogenesis and tumorigenesis [19,29,30]. This evidence suggests that the IDH-mutant-related decrease of α-KG stabilizes HIF-1α and leads to aberrant cellular proliferation. IDH mutations are prevalent in several rare malignancies, such as iCCA, glioma, acute myeloid leukemia, chondrosarcoma, thyroid carcinoma, angioimmunoblastic T-cell lymphoma, and other cancers [31,32,33,34,35,36]. 

## 4. Targeting IDH in Cancer

Based on this biological rationale, many research efforts have been established to identify IDH-directed therapies and investigate them as potential anti-cancer drugs. Initial studies have shown the in vitro efficacy of IDH inhibitors. In 2012, Popovici-Muller et al. developed an IDH1 inhibitor (AGI-5198) that provided up to 90% reduction of 2-HG in a U87 glioblastoma xenograft mouse model [37]. Rohle et al. confirmed the efficacy of AG-5198 in inhibiting 2-HG production in patient-derived glioma xenografts, also showing that it promoted the expression of markers for differentiation and decreased cellular proliferation and histone methylation in the same cell line [38]. However, because of its suboptimal pharmacodynamic profile with a rapid metabolism and clearance, the advancement in clinical trials of AGI-5198 has been precluded [39]. Another IDH1-mutant inhibitor (BAY1436032) was tested in two preclinical experiments using different dosing regimens to treat IDH1-mutated intracranial xenografts in BALB/c nude mice [40]. While a 150 mg/Kg daily dose did not significantly reduce the size of intracranial xenografts, a significant decrease in intratumoral D2HG and a statistically significant increase in animal survival was found in the treated group. In the second experiment, investigators found that a twice-daily 70 mg/kg dose of the drug prolonged animal survival. Two dose escalation and expansion phase I trials for both acute myeloid leukemia (AML) and solid tumors (including glioma) are currently ongoing (ClinicalTrials.gov Identifier: NCT02746081). The IDH1 inhibitor IDH305 has also shown significant 2HG reduction in IDH1-mutant colorectal cancer cell lines and substantial brain penetrance in murine models [41]. It has been tested in humans with IDH-mutant glioma, AML, and other solid tumors, and the first phase 1 first results in safety data are promising [42]. To improve the pharmacodynamic profile of AGI-5198, the IDH1 inhibitor AG-120 (ivosidenib) was developed. Testing in animal models with an intact blood-brain barrier showed that it had a low level of brain penetration [37]. However, its ability to modulate the oncogenic properties of cancer cells and reducing 2HG has been demonstrated by the induction of cellular differentiation in AML myeloblasts and through the inhibition of cell migration and invasion in a chondrosarcoma cell line [43,44,45]. A phase I clinical trial showed promising results in objective response and a favorable safety profile, making ivosidenib an orphan drug for glioma in 2018, and leading to its approval by the FDA in July 2018 for adults with refractory or relapsed AML [46]. Furthermore, this drug is currently under evaluation in many clinical trials, both in hematologic and solid tumors. The first IDH2 inhibitor developed was AGI-6780, which induced thex differentiation of IDH2 mutated erythroleukemia and primary human AML cells [47]. Unfortunately, further clinical development was blocked by the lack of in vivo evidence and the subsequent development of IDH2 inhibitor enasidenib. Yen et al. evaluated the efficacy of the IDH2 inhibitor AG-221 (enasidenib) in an IDH2-mutant AML xenograft mouse model, showing a significant decrease in the marrow, plasma, and urine 2-HG, along with a dose-dependent survival benefit [48]. On this basis, the first-in-human phase 1 clinical trial investigated the safety and tolerability of AG-221 in patients with relapsed or refractory IDH2-mutant AML [49]. Selecting a 100 mg/die dose based on the results of a phase 1 dose-escalation study, around 40% of patients had an objective response, including 20% with a complete response. Treatment was well-tolerated, with the most common adverse events being nausea, diarrhea, fatigue, and fever. Moreover, enasidenib induced a considerable decrease in plasma 2-HG levels in most of the patients treated. This result led to the FDA approval of enasidenib for relapsed or refractory IDH2-mutant AML in 2017 [50], and also to the development of other trials evaluating this drug in various AML subpopulations and advanced solid tumors. Finally, the pan-IDH inhibitor AG-881 (vorasidenib), an oral inhibitor of both IDH1 and IDH2-mutant, has been evaluated for use in IDH-mutant solid and hematologic malignancies [51]. It was demonstrated that ex vivo treatment of primary human AML blasts with AG-881 induced myeloid differentiation. It was also shown to fully penetrate the blood-brain barrier, implicating its potential role in treating both IDH-mutant AML and glioma patients. For this reason, two multicenter clinical trials investigating AG-881 in solid tumors and hematologic malignancies, respectively, are currently ongoing [52,53].

Despite the promising data described, initial evidence on the mechanisms of acquired resistance to these small molecule inhibitors have been reported, resulting in progressive disease with an increase in plasma 2-HG concentration. In a first report, two patients with IDH2-mutant AML developed resistance to the mutant IDH2 inhibitor enasidenib as a result of the emergence of second-site IDH2 mutations in trans (Q316E, I319M) in the wild-type allele. This cooperated with the gain-of-function mutation (R140Q) on the other allele in inducing resistance either by breaking up the hydrogen bond between the IDH2 dimer and enasidenib or by hindrance of binding of the IDH2 dimer to enasidenib [54]. Similarly, mechanisms of resistance to ivosidenib have been described. Receptor tyrosine kinase (RTK) pathway mutations have been associated with primary resistance to this drug, while multiple mechanisms contributed to acquired resistance, such as the development of RTK pathway mutations and 2-HG restoring mutations. Furthermore, multiple concurrent mechanisms have been identified in single patients [55].

## 5. IDH Mutations in Cholangiocarcinoma

The discovery of mutations in IDH isoforms (IDH1 and IDH2) has been a major breakthrough in the translational research of cholangiocarcinoma. They have been reported to occur in approximately 15−20% of iCCA [56], while rarer evidence of these molecular alterations is present for both eCCA and GBC [33,57]. IDH1 mutations are more common than mutations of IDH2, with IDH1 hotspots located in the arginine 132 residue, IDH1-R132C (44%), and IDH1-R132G (14%) [56]. The prognostic value of IDH mutations in cholangiocarcinoma remains controversial.

As previously mentioned, these mutations cause elevated levels of the oncometabolite 2-hydroxyglutarate (2-HG), which can be detected in tissue and blood as a surrogate biomarker for IDH-mutant iCCA [57]. Elevation of 2-HG is associated with higher DNA CpG methylation and altered histone methylation. Epigenetic changes cause blocks in the cellular differentiation of iCCA cells. Moreover, IDH mutations cause alterations in the hypoxia signaling, collagen processing, and activation of EMT via increased expression of ZEB1 and decreased levels of miR-200. In addition, IDH1/2 mutations often interact with TK and MAPK-dependent signaling pathways [56]. Indeed, iCCA cells often have higher levels of total ERK 1 and 2, phospho-ERK 1 and 2, and a downstream target, phospho-CREB [58]. IDH1 and IDH2 mutations are mutually exclusive with NRAS/KRAS and FGFR mutations and may co-exist with BAP1 mutations [56]. A recent series analyzed 149 tumor samples of ctDNA from 104 patients. IDH1 mutations were found in 19.1% of cases. In particular, 17 patients had IDH1 mutations, with 70.6% showing IDH1-R132C, 23.5% showing IDH1-R132L, and 5.9% showing IDH1-R132G. Concordance of findings for paired tissues and ctDNA was complete (100%). Therefore, in cases of insufficient tumor tissue for molecular analysis, ctDNA-based approaches may be used instead and allow the detection of known mutations as well. 

## 6. Clinical Development of IDH Inhibitors in Cholangiocarcinoma

As discussed earlier in the text, multiple IDH-selective inhibitors have been developed so far. AG-120 (ivosidenib, Agios) is the most developed IDH inhibitor for cholangiocarcinoma patients. A cohort of 168 patients with IDH-1 mutated cholangiocarcinoma, chondrosarcoma, and glioma were treated with ivosidenib in order to evaluate pharmacokinetic and pharmacodynamics profiles. The drug demonstrated a good oral exposure at the ideal oral dose of 500 mg qd. Moreover, its half-life was prolonged (mean 40−102 hours after a single dose). After multiple doses, the plasmatic levels of 2-HG were reduced by up to 98% of normal values recorded in healthy controls. In a phase I dose-escalation and expansion trial, 73 patients with IDH-1 mutations received AG-120. No dose-limiting toxicities were reported, and a dose of 500 mg qd was selected for the expansion cohort. The most common recorded toxicities were fatigue (42%), nausea (34%), diarrhea (32%), abdominal pain (27%), and vomiting (23%). Grade ≥3 adverse events included fatigue (3%) and hypophosphatemia (1%). A total of 6% of patients achieved a partial response, with 56% experiencing a stable disease [59,60]. Median PFS was 3.8 months (95% CI 3.6−7.3) [61]. The recent phase III ClarIDHy trial randomized 230 patients with advanced pretreated IDH-1 mutated cholangiocarcinoma to ivosidenib or a placebo (2:1). Ivosidenib/placebo was given at the dose of 500 mg qd in 28-days cycles. Crossover to ivosidenib was permitted on radiological progression during the placebo. Median follow-up for PFS was 6.9 months (interquartile range 2.8−10.9). Ivosidenib showed an advantage in median PFS compared to the placebo (2.7 vs 1.4 months, HR 0.37, 95% CI 0.25.0.54, *p* < 0.0001) [10]. The PFS rates at 6 and 12 months were superior for ivosidenib (32 and 21.9% vs 0% at both 6 and 12 months for placebo arm, respectively). The median OS was significantly longer in the experimental arm, with 10.8 months for ivosidenib and 6 months for the placebo (HR 0.46, *p* = 0.0008) after adjustment for crossover. The response rate was 2.4% [61], and the most common recorded averse event in both treatment groups was ascites (4/59, 7% for placebo and 9/121, 7% for ivosidenib) [10].

Many phase I and II trials are currently testing IDH1/2 inhibitors in cholangiocarcinoma and are listed in Table 1. Among these compounds, FT-2102 (olutasidenib) is very promising. It is a brain-penetrant IDH1-mutant inhibitor that reduces 2-HG production in xenograft IDH1^R132H^ in vivo models and has a good cell permeability [62]. It has been tested in ongoing phase I/II clinical trials in patients with IDH1-mutated relapsed, refractory AML, and myelodysplastic syndrome (MDS), both in monotherapy and combination with azacitidine. At present, FT-2102 is also under investigation for advanced solid tumors and gliomas (NCT03684811). This trial is planned to be structured in two parts: patients affected by iCCA will be enrolled in a phase I/II trial for dose determination, and the clinical activity will be evaluated in the single-agent arm and/or in the gemcitabine/cisplatin combination arm. 

Beyond selective IDH inhibitors, dasatinib, a multi-tyrosine kinase, was shown to be active against the growth of IDH-mutant iCCA cells. Indeed, SRC is a target of dasatinib, and IDH mutated iCCA cells are critically dependent on SRC activity for survival and proliferation [63]. Therefore, its use is being explored in a phase II trial with patients affected by advanced iCCA (NCT02428855). The recruitment for this trial was completed, and the results are awaited. 

## 7. Future Perspectives and Conclusions

Biliary tract cancer remains a challenging disease as recurrence rates are high after surgery, and chemotherapy has a limited efficacy both in adjuvant and advanced settings. However, mounting evidence is demonstrating a clinically meaningful advantage for a molecularly selected subset of BTC treated with novel targeted therapies. Central to this is the IDH1 inhibitor ivosidenib, which has been the first targeted agent to show a survival benefit in a randomized phase III trial of IDH1-mutant cholangiocarcinoma patients. Beyond cytotoxic chemotherapy, this practice-changing approach is paving the way for the personalized oncology era in BTC. Some challenges are still ahead of us: more advanced multiplatform analyses are warranted to enhance the detection of novel actionable molecular vulnerabilities in patients without exploitable alterations (roughly 50%). Primary and acquired resistance to IDH1 inhibitors is also emerging, which results in treatment failure. Finally, the implementation of multinational collaborative efforts along with next-generation clinical trials using expansion platform design is desirable to better study this relatively rare disease characterized by low-prevalence molecular hallmarks. In conclusion, though BTC research has historically lagged behind other cancer types, precision oncology has begun to realize the potential of this hard-to-treat tumor by changing conventional treatment algorithms, particularly in intrahepatic cholangiocarcinoma.

## Figures and Tables

**Figure 1 cancers-12-03310-f001:**
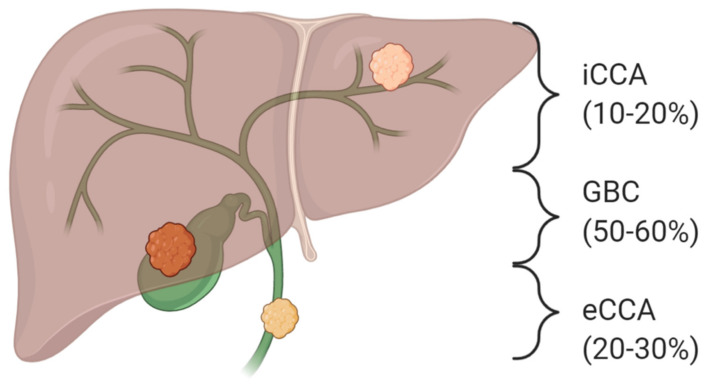
Anatomical sub-classification of biliary tract cancers. Based on the anatomical site of origin within the biliary tree, biliary tract cancers are subdivided into intrahepatic (iCCA) and extrahepatic cholangiocarcinoma (eCCA) and gallbladder carcinoma (GBC).

**Figure 2 cancers-12-03310-f002:**
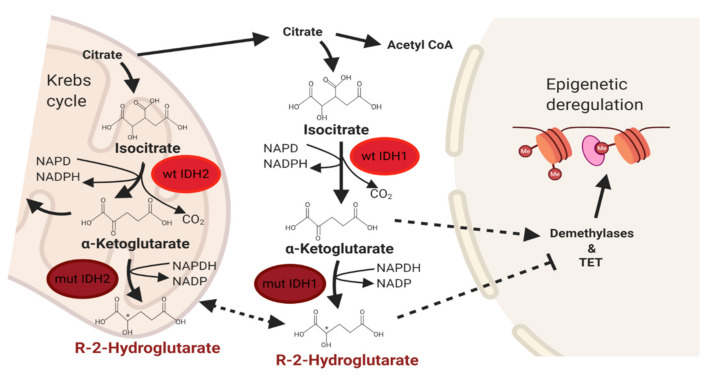
Normal and deregulated isocitrate dehydrogenase (IDH) signaling in cancer. Abbreviations: wt IDH1/2, wild-type isocitrate dehydrogenases 1 and 2; mut IDH1/2, mutant isocitrate dehydrogenases 1 and 2; TET, ten-eleven translocation.

**Table 1 cancers-12-03310-t001:** Selected clinical trials of IDH1/2 inhibitors in advanced cancers including cholangiocarcinoma.

Trial Number	Compound	Phase	Setting	Status
NCT02428855	Dasatinib	II	Advanced iCCA IDH1/2 mut	Completed
NCT04088188	Gem/Cis + ivosidenib or pemigatinib	I	Advanced CCA	Not yet recruiting
NCT03684811	FT2102	Ib/II	Advanced iCCA IDH1 mut	Active, not recruiting
NCT02273739	Enasidenib	I/II	Advanced iCCA IDH2 mut	Completed
NCT02381886	IDH305	I	Advanced tumours IDH1 R132 mut	Active, not recruiting
NCT02481154	AG-881	I	Advanced tumours IDH1/2 mut	Active, not recruiting
NCT04056910	Ivosidenib + nivolumab	II	Advanced tumours IDH1 mut	Not yet recruiting
NCT04521686	LY3410738	I	Advanced tumours IDH1 R132 mut	Recruiting
NCT02746081	BAY 1436032	II	Advanced tumours IDH1 R132X *mut*	Active, not recruiting
NCT02073994	AG-120	I	Advanced tumours IDH1 mut	Active, not recruiting
NCT03878095	Olaparib + ceralasertib	II	Advanced tumours IDH1/2 mut	Recruiting

CCA: cholangiocarcinoma; iCCA: intrahepatic cholangiocarcinoma; IDH: isocitrate dehydrogenase; mut: mutated. Gem: gemcitabine; Cis: cisplatin.

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
