# Peer review of "IDH Signalling Pathway in Cholangiocarcinoma: From Biological Rationale to Therapeutic Targeting"

_cancers, 2020, doi:10.3390/cancers12113310_

Round 1

Reviewer 1 Report

Comments to Authors:

Title: IDH signalling pathway in cholangiocarcinoma: from biological 

rationale to therapeutic targeting.

This paper describes IDH 1/ IDH 1 pathway as well as IDH inhibitors in cholangiocarcinoma from biological rationale to therapeutic targeting.

This paper including relatively large number of references so the results may be highly reliable.

But there are some questions and the author is requested to add the descriptions according to comments as below.

Minor points

  • Anatomical sub-classification of biliary tract cancers

In figure 1, the author described about classification of biliary tract cancers.

The author should add the descriptions of classification of cholangiocarcinoma

2) IDH inhibitors in cholangiocarcinoma

There is no detailed description about other IDH1/2 inhibitors except AG-120.

The author should add the descriptions about other compounds such as FT2102 and BAY-1436032.

Author Response

We really appreciated Reviewer's n.1 comments and suggestions. 

  1. We added directly in the text the following statement:"[further divided into perihilar (pCCA) and distal cholangiocarcinoma (dCCA)]".
  2. The following paragraph was added to the text: "Among these compounds, FT-2102 (Olutasidenib) is very promising. It is a brain-penetrant IDH1-mutant inhibitor, which reduces 2-HG production in xenograft IDH1R132H in vivo models and has a good cell permeability [63]. It has been tested in an ongoing phase I/II clinical trials in patients with IDH1-mutated relapsed, refractory AML and myelodysplastic syndrome (MDS) both in monotherapy and combination with Azacitidine. At present, FT-2102 is also under investigation for advanced solid tumors and gliomas (NCT03684811). This trial is planned to be structured in two parts: patients affected by iCCA will be enrolled in a phase I/II trial for dose determination and the clinical activity will be evaluated in the single-agent arm and/or in the gemcitabine/cisplatin combination arm". 

Reviewer 2 Report

The present review from Salati et al. is an interesting and well-written summary of the relevance of IDH mutations in cholangiocarcinoma. Authors describe the potential of these mutations to be used as targets in therapy, and summarize the main drugs used as IDH inhibitors to treat cholangiocarcinoma patients. Only a few minor issues should be addressed: 

I suggest to include a section (or at least comment) the potential mechanisms of Ivosidenib resistance. In the last section (Future perspectives and conclusions) authors explain that “primary and acquired resistance to IDH1 inhibitors is emerging that results in treatment failure”. If authors include some information about the main mechanisms of chemoresistance involved in ivosidenib (or other IDH inhibitor) resistance could improve the quality of this review.

Author Response

We thank Reviewer n.2 for the comments and suggestions provided.

In order to reply to them, the following paragraph was added to the text: 

"Despite the promising data described, initial evidence on mechanisms of acquired resistance to these small molecule inhibitors have been reported, resulting in progressive disease with increase in plasma 2-HG concentration. In a first report, two patients with IDH2-mutant AML developed resistance to the mutant IDH2 inhibitor enasidenib as a result of the emergence of second-site IDH2 mutations in trans (Q316E, I319M) in the wild-type allele. This cooperates with the gain-of-function mutation (R140Q) on the other allele in inducing resistance either by breaking up the hydrogen bond between the IDH2 dimer and enasidenib or by hindrance of binding of the IDH2 dimer to enasidenib [54]. Similarly, mechanisms of resistance to Ivosidenib have been described. Receptor tyrosine kinase (RTK) pathway mutations have been associated with primary resistance to this drug, while multiple mechanisms contributed to the acquired resistance, such as the development of RTK pathway mutations and 2-HG restoring mutations. Furthermore, multiple concurrent mechanisms have been identified in single patients [55]".

Reviewer 3 Report

The authors summarized in detail the current state of research regarding IDH mutations in cholangiocarcinoma. The review is well written, and it provides details that would be interesting for pre-clinical and clinical researchers alike.

More than to ask an addendum, I would like just to suggest the authors to discuss the following idea, but only if they agree that there is enough information available to do it in a suitable way:

My impression from this review and data collected in other cancers, is that the impact of IDH mutations on cancer development is mainly explained by epigenetic modifications. There may be other mechanisms to unveil, but so far this is the main mechanism postulated. These epigenetic alterations seem to have an impact early in the oncogenic development. Since there has been somehow disappointing results in cholangiocarcinoma and other cancers using IDH inhibitors (except that phase III trial using Ivosidenib) and, as the authors described, the prognostic value of IDH mutations of cholangiocarcinoma is rather controversial, do the authors think mutated IDH is a is a target correctly validated as a cancer vulnerability? Is there enough evidence to support the claim that depletion of 2HG is an efficient way to reduce cancer burden? How much is a tumor dependent on high levels of 2HG to sustain malignancy? How likely is that IDH mutations had an impact on early carcinogenesis, but once the malignancy increased, the tumor relies less on IDH mutations to sustain invasion and metastasis. 

This discussion is not mandatory, but it may add more value to the review and contribute to future of IDH mutations as cancer target.

Author Response

We thank Reviewer n.3 for the enlightening suggestion regarding a critical appraisal on the role of IDH mutations as a driver and target in cancer.

However, the data to discuss this topic are largely lacking in CCA.

Besides, our opinion is that this probably deserves an extensive discussion in a dedicated review article.